# AlveoMPU: Bridging the Gap in Lung Model Interactions Using a Novel Alveolar Bilayer Film

**DOI:** 10.3390/polym16111486

**Published:** 2024-05-23

**Authors:** Minoru Hirano, Kosuke Iwata, Yuri Yamada, Yasuhiko Shinoda, Masateru Yamazaki, Sayaka Hino, Aya Ikeda, Akiko Shimizu, Shuhei Otsuka, Hiroyuki Nakagawa, Yoshihide Watanabe

**Affiliations:** 1Frontier Research Management Office, Toyota Central R&D Labs., Inc., 41-1 Yokomichi, Nagakute 480-1192, Aichi, Japan; e4610@mosk.tytlabs.co.jp (Y.Y.); ywatanabe@mosk.tytlabs.co.jp (Y.W.); 2Organic Device Development Department, Material Development Division, Toyoda Gosei Co., Ltd., 1-1 Higashitakasuka, Futatsudera, Ama 490-1207, Aichi, Japan; kosuke.iwata.2@toyoda-gosei.co.jp (K.I.); masateru.yamazaki@toyoda-gosei.co.jp (M.Y.); sayaka.hino@ts.toyoda-gosei.co.jp (S.H.); aya.ikeda@toyoda-gosei.co.jp (A.I.); akiko.shimizu@toyoda-gosei.co.jp (A.S.); shuhei.otsuka@ts.toyoda-gosei.co.jp (S.O.); hiroyuki.nakagawa@toyoda-gosei.co.jp (H.N.)

**Keywords:** alveolar, polyurethane, porous membrane, in vitro, drug screening

## Abstract

The alveoli, critical sites for gas exchange in the lungs, comprise alveolar epithelial cells and pulmonary capillary endothelial cells. Traditional experimental models rely on porous polyethylene terephthalate or polycarbonate membranes, which restrict direct cell-to-cell contact. To address this limitation, we developed AlveoMPU, a new foam-based mortar-like polyurethane-formed alveolar model that facilitates direct cell–cell interactions. AlveoMPU features a unique anisotropic mortar-shaped configuration with larger pores at the top and smaller pores at the bottom, allowing the alveolar epithelial cells to gradually extend toward the bottom. The underside of the film is remarkably thin, enabling seeded pulmonary microvascular endothelial cells to interact with alveolar epithelial cells. Using AlveoMPU, it is possible to construct a bilayer structure mimicking the alveoli, potentially serving as a model that accurately simulates the actual alveoli. This innovative model can be utilized as a drug-screening tool for measuring transepithelial electrical resistance, assessing substance permeability, observing cytokine secretion during inflammation, and evaluating drug efficacy and pharmacokinetics.

## 1. Introduction

The lungs, essential for sustaining life by facilitating gas exchange, may also serve as sites for inflammation and respiratory disease reactions induced by inhaled environmental chemicals and allergens [1,2,3]. Thus, accurately replicating the alveolar structure and function in vitro is crucial for understanding the pathophysiology of lung diseases and developing new treatments or drug screening tools. The alveoli consist of closely packed type I and II alveolar epithelial cells and pulmonary microvascular endothelial cells, which form a dual-layered sheet with a total thickness of approximately 2 μm, functioning as independent sheets that facilitate gas exchange [4]. Thin-film substrates, such as cell culture inserts that culture two different cell types separately, are typically used to simulate this microenvironment [5,6]. In these previous studies, a bilayer alveolar model was constructed using commercial well-plate format cell culture inserts. In this model, the primary cultured cells of alveolar epithelial cells and capillary endothelial cells, which are the major cell types constituting the alveoli, as well as cell lines such as cancer cells, are separated by a porous membrane. This model has been utilized to study the effects of inflammatory substances, evaluate drug efficacy, and assess pharmacokinetics. Huh and colleagues developed an elaborate 3D alveolar model using micro electro mechanical systems (MEMS) technology by creating a 10 µm thick PDMS-based porous membrane to replicate the bilayer alveolar sheet. They also demonstrated that this bilayer sheet could be cultured under mechanical stretching induced by pneumatic pressure [7]. However, MEMS technology is not readily accessible to the general research community, and as the complexity of MEMS structures increases, issues such as reproducibility between samples arise, making it not necessarily suitable for high-throughput assays like drug screening. However, recent studies have reported the formation of poly(caprolactone) PCL fibers with nanometer-scale diameters using electrospinning. For example, an alveolar model partitioned by a fibrous thin film with a thickness of 2 µm has been constructed [8].

Nevertheless, commercially available cell culture inserts made from materials such as polyethylene terephthalate (PET) or polycarbonate (PC) are linearly perforated by electron beams [9]. The pore size may restrict cell movement or allow infiltration. Therefore, these inserts are unable to stably maintain a bilayer structure of two cell types in a direct or proximate arrangement. Moreover, their membrane thickness of approximately 10 µm poses a challenge, limiting the interaction between the two cell layers (Appendix A). Theoretically, minimizing the membrane thickness can resolve this issue. However, maintaining the stability of the ultra-thin membrane structure during culture remains challenging. Additionally, while the fibrous thin films produced by the electrospinning of PCL have achieved sufficient thinness compared to that of conventional films, they continue to have a partition distance of 2 µm. Furthermore, the strength characteristics of these fibers are not addressed in the article.

To ensure the strength of the thin film and enhance the interaction between cells seeded on both sides of the membrane, it is necessary to form a framework structure for strength reinforcement and locally thin regions on the membrane surface. Foamed thin films are optimal for this purpose, where non-foamed areas function as relatively strong films, while foamed areas form a thin skin layer on the membrane surface. This could provide an interaction field for cells on both sides of the membrane once openings are created. Methods for controlling polymer foaming include the breath figure method and chemical foaming. The former generates a honeycomb structure with densely arranged pores across the membrane surface, resulting in a large area ratio of foamed areas [10,11,12]. Furthermore, the foam has large isotropic spherical voids within the membrane, making it difficult for cells on both sides of the membrane to form a bilayer thin sheet separated by an ultra-thin membrane owing to the thickness of the membrane and the voids inside the membrane. Conversely, chemical foaming tends to yield a more random and sparse structure, allowing flexibility in adjusting material properties, foam diameter, and density [13,14]. Polyurethane is synthesized by the addition polymerization of polyols and polyisocyanates and is known as a biocompatible material, used as a film for cultured cell substrates and wound healing materials [15,16]. Although there are concerns about the potential toxicity of unreacted substances and residual solvents in these materials, the technology to safely manufacture polyurethane is well established. As a result, polyurethane is already applied in various products that come into direct contact with humans, such as automotive parts, clothing, stretchable sportswear, shoe insoles that require cushioning, sponges, and highly safe items like medical catheters [17]. Attempts have been made to adjust physical properties and improve cell adhesiveness by mixing other materials with polyurethane [18,19]. However, meeting the criteria for in vitro alveolar models, specifically having the adequate strength for conventional cultured cells and featuring areas with thin layers that enable two layers of cell sheets separated by a membrane to be in close proximity, has been challenging. 

Therefore, we employed chemical foaming of polyurethane to prepare the thin film of the cell culture substrate. By introducing steam and heat during the polymerization process, it was possible to control the foam size. This allowed us to fabricate a porous polyurethane material (P-PU) with a mortar-like structure that has anisotropic pores that are larger at the top and smaller near the mold interface [20]. Through detailed analysis of the surface morphology and physical properties of the obtained material, we characterized it as a cultured cell substrate and obtained basic data on cell extension and adhesion. Moreover, when cells constituting the alveoli were seeded on this P-PU and evaluated as an alveolar model, it demonstrated potential for inflammation assessment and pharmacokinetics analysis. We named this alveolar model based on P-PU AlveoMPU (Figure 1). This model can provide data closer to the in vivo conditions for lung disease research and in vitro drug efficacy testing. AlveoMPU is expected to bridge the gap between in vitro testing and biological systems, fostering innovation in lung pathophysiological research and treatment development (Appendix A).

## 2. Materials and Methods

### 2.1. Preparation of Foamed Porous Polyurethane Film

The P-PU used in this study was prepared in accordance with the procedure described in US PATENT: 9637722 [20]. Briefly, 100 volumes of polyol, 18 volumes of diethylene glycol, and 7 volumes of water were added to 267 volumes of tetrahydrofuran, and the mixture was stirred to obtain a uniform mixture. Subsequently, the mixture was cooled while stirring. Isocyanate was added to the cooled mixture and the reaction mixture was then transferred onto a polypropylene (PP) plate and spread uniformly using a spin coater to form a thin film. The spin-coated PP plate was subsequently placed in a chamber at 40 °C and 95% humidity, where the water vapor present in the film reacted with isocyanate to obtain a foamed P-PU film. Subsequently, a curing acceleration reaction was carried out for 12 h at 60 °C. By removing the membrane from a 24-well cell culture insert and replacing it with the P-PU film, we obtained a cellular scaffold for AlveoMPU. Hereafter, P-PU is defined as the film itself and AlveoMPU is defined as a composite of P-PU with cells and the cell culture insert.

### 2.2. Electron Microscopy

Sample preparation and electron microscopy observations were outsourced to Hanaichi UltraStructure Research Institute (Okazaki, Japan). Briefly, cell samples for transmission electron microscopy (TEM) analysis were fixed in 0.1 M phosphate-buffered 2% glutaraldehyde solution, followed by post-fixation in 2% osmium tetroxide for 2 h in a chilled ice bath. Subsequently, the specimens were dehydrated using progressively concentrated ethanol solutions before being encapsulated in an epoxy resin. Ultra-thin slices (80–90 nm) were obtained using an ultramicrotome. These sections were stained with uranyl acetate for 15 min, subsequently with a lead-astaining solution for 2 min, and examined by TEM at an acceleration voltage of 100 kV (H-7600; Hitachi, Chiyoda, Japan). 

For scanning electron microscopy (SEM) observation, samples were initially stabilized in a 0.1 M solution of phosphate-buffered 2% glutaraldehyde, followed by further stabilization in 2% osmium tetroxide for 2 h in a chilled ice bath. Subsequently, the samples were dehydrated using a series of progressively concentrated ethanol concentrations and desiccated by freeze-drying in t-butyl alcohol. After drying, the samples were coated with osmium using a plasma ion coater and subsequently analyzed by SEM at an operating voltage of 5 kV using a JSM-7500F (Hitachi).

### 2.3. Atomic Force Microscopy

The surfaces of the P-PU used in AlveoMPU were characterized by performing atomic force microscopy (AFM) using a NanoNavi E-sweep scanning probe microscope (Hitachi). In the evaluation of membrane smoothness, the arithmetic (mean) average roughness (Ra), which describes a deviation of a surface from a theoretical center line, was used [21]. Detailed experimental conditions and discussions can be found in Appendix A.

### 2.4. Pore Size Measuring with Laser Microscopy Observations

The P-PU film was carefully peeled from the PP plate. Before peeling, a utility knife was used to score the film in a crosshatch pattern to facilitate easier removal. Subsequently, the front and back surfaces of the film were meticulously measured using a digital microscope (Keyence Corporation, Higashi-Yodogawa, Japan). To adequately capture the variance in pore size, larger pores were observed at a magnification of 400×, while smaller pores were examined at a higher magnification of 500×.

### 2.5. Measurement of Dynamic Contact Angle

Since the culture medium was an aqueous-based liquid, the water contact angle on the AlveoMPU membrane was evaluated using a DMo-501 instrument (Kyowa Interface Science Co., Ltd., Shinza, Japan) under controlled condition at 21.0 °C. For static contact angle measurements, a water droplet of 0.5 µL was automatically dispensed on five different spots of the film using a 28-gauge syringe, and the arithmetic mean was calculated. The behavior of water droplets on the film surface as a function of time was also investigated to determine dynamic contact angle. A 0.5 µL drop of water was automatically spotted on the film surface and the wetting front was monitored every 5 s. To ensure reproducibility, measurements were repeated on different spots on the film. The image of the sessile drop was captured every 5 s, and the obtained images were analyzed using the axisymmetric drop shape analysis technique [22] using the software provided with the device. For comparison, identical measurements were conducted on a flat polyurethane film lacking a porous structure (referred to as Flat-PU: F-PU). 

### 2.6. Tensile Testing (Stress–Strain Curves)

Tensile tests were performed using a compression-testing machine (AG-XPLUS 500N; Shimadzu Corporation, Nishinokyo, Japan). Type 7 dumbbell specimens were selected for testing at a set tensile speed of 100 mm/min and a load cell capacity of 5 N. The measurement commenced with die cutting the film into the shape of a Type 7 dumbbell to create test specimens. The thickness of each specimen was measured and recorded. The specimens were then affixed to a tension-specific support paper, clamped onto the device, and subjected to tensile testing until breakage. The stress–strain curve was derived from the obtained test force and elongation data, and the elastic modulus was calculated from the slope of the strain range of 1–3%.

### 2.7. Cell Culture and AlveoMPU Preparation

The steps for constructing AlveoMPU are shown in Appendix A. The cell culture inserts and porous PU films were first disinfected by immersion in CIDEX OPA (World Precision Instruments, Chiyoda, Japan), followed by washing with sterile ultrapure water, drying, and sterilization with UV light. Subsequently, a collagen gel culture kit (FUJIFILM Wako Pure Chemical Corporation, Osaka, Japan) was used according to the manufacturer’s protocol, and collagen was applied to the bottom of the insert, with excess quantities promptly aspirated to achieve a thin coating. Gelation was promoted by incubation in a CO_2_ incubator for 30 min.

To form a monolayer of pulmonary alveolar epithelial cells, human alveolar epithelial cells (HAEpiCs; CellBiologics, Chicago, IL, USA) were seeded onto the inserts at a density of 1.0 × 10^5^ cells/cm^2^ (volume: 2 µL; using Pulmonary Alveolar Epithelial Cell Medium; ScienCell Research Laboratories), and the plate side was filled with 750 µL medium. The next day, the medium was switched to a growth factor-reduced medium (FBS 2.5%, EpiCGS-free), and the medium was replaced every 2 days.

For the formation of a bilayer sheet of HAEpiC and human lung microvascular endothelial cells (HLMVECs; Lonza, Sagamihara, Japan), after coating, a silicon ring was attached to the insert, and HLMVECs were seeded on the insert at a density of 1.0 × 10^5^ cells/cm^2^ (volume: 400 µL; cultivation area: 0.67 cm^2^; using EGM™-2MV BulletKit™; Lonza) and incubated for 1 h. Subsequently, the culture medium was removed, the silicon ring was detached, and the insert was placed in a 24-well plate. Next, 200 µL vascular culture medium was added into the insert, 750 µL medium was added to the plate side, and the cells were cultured for 1 day. Pulmonary alveolar epithelial cells were seeded at a density of 1.0 × 10^5^ cells/cm^2^ (volume: 200 µL; using complete HAEpiC medium) and 750 µL vascular culture medium was added to the plate well followed by 1 day culture. The medium inside the insert was changed to growth factor-reduced medium (200 µL/well, the medium without EpiCGS supplement, with fetal bovine serum reduced from 2% to 0.5%) and 750 µL of endothelial culture medium was added to the plate well, with the medium being changed every 2 days. During the barrier formation process, electrical resistance of the cell layer was measured daily from the first day of bilayer formation to the seventh day using an EVOM2-STX3 device.

### 2.8. Immunostaining Cells on AlveoMPU

Cells cultured on AlveoMPU were fixed with 4% paraformaldehyde (PFA) for 15 min and washed with phosphate-buffered saline (PBS). For the cross-sectional evaluation, sample preprocessing and sectioning were outsourced to Biopathology Institute Co., Ltd. (Kunisaki, Japan). Next, the deparaffinized sections were immersed in citrate buffer (pH 6.0) for antigen retrieval. They were then heated in an autoclave at 120 °C for 10 min and naturally cooled to room temperature, followed by washing with tris-buffered saline (TBS). For immunostaining of standard planar cultures, after fixation with PFA, permeabilization was performed using 0.1% Triton-X100 (Nacalai Tesque, Nakagyo, Japan). The procedures for primary and secondary antibody treatment (Appendix A) remained consistent for both planar cultures and cross-sectional evaluations. Primary antibodies were diluted in Can Get Signal^®^ immunostain Solution B (Toyobo, Osaka, Japan) and incubated at 4 °C for 1 h, followed by washing with TBS. Secondary antibodies were diluted in Can Get Signal B and applied to the samples. If DAPI staining was required, it was conducted at this stage. After incubation with secondary antibodies at room temperature for 1 h, samples were washed with TBS and observed under a fluorescence microscope. Plasma membrane staining was performed by applying PBS containing DAPI and PlasMem Bright Red (Dojindo Laboratories, Kamimashiki, Japan) after primary antibody staining. Samples were then incubated in a 37 °C, 5% CO_2_ incubator for 10 min, followed by washing with TBS and fluorescence microscopy observation.

### 2.9. Intervention of AlveoMPU Using Proinflammatory and Anti-Inflammatory Substances and Permeability Assay

For quantifying cytokine secretion from the AlveoMPU model, culture media were prepared and collected as follows: the insert side was supplemented with pulmonary epithelial medium containing either 100 nM fluticasone propionate (#F9428; Sigma-Aldrich, Meguro, Japan) or dimethyl sulfoxide (CultureSure^®^ DMSO #031-24051; FUJIFILM Wako Chemicals), and the plate side was supplemented with fresh endothelial media. Subsequently, the plates were incubated in a CO_2_ incubator for 1 h. Following incubation, media alone, media containing 10 µg/mL high molecular weight polyinosine-polycytidylic acid (Poly [I:C]; #tlrl-pic; InvivoGen, San Diego, CA, USA), and pulmonary epithelial medium containing 10 µg/mL lipopolysaccharide (LPS) from *Escherichia coli* 0111:B4 (ELPS; InvivoGen) were applied to the insert side, with fresh vascular media added to the plate side. The plates were incubated in a 5% CO_2_ environment for 24 h. Subsequently, the media from both the insert and plate sides were collected and used as samples for cytokine quantification. Enzyme-linked immunosorbent assay (ELISA) kits for IL-6 (#HS600C), TNFα (#HSTA00E) were purchased from R&D Systems. An IL-8 ELISA kit (#KE00006) was purchased from Proteintech (Koutou, Japan). Cytokine quantification was performed following the manufacturers’ instructions.

For permeability studies, samples were collected as follows: the insert side was supplemented with medium-containing fluorescein-labeled dextran with a molecular weight of 4k Da FITC-Dextran (Sigma-Aldrich, #FD4, Meguro, Japan), and fresh medium was added to the plate side to initiate the permeability test. Samples were incubated in a CO_2_ incubator for 24 h, after which the medium from the plate side was collected, and the fluorescence intensity was measured using a SpectraMax iD3 plate reader (Molecular Devices, Chuo, Japan). Fluticasone was obtained from the Tokai Technical Center. The apparent permeability coefficient (Papp) was calculated from the concentrations of fluorescent dextran and fluticasone, obtained using a previously described method [23].

### 2.10. Statistical Analysis

Statistical analyses were conducted using GraphPad Prism software (version 6.03J). For comparison of mean values among groups, analysis of variance (ANOVA) was employed. Where significant differences were identified by ANOVA, post-hoc multiple comparisons were performed using Dunnett’s test to evaluate differences between control and experimental groups. A *p*-value of less than 0.05 was considered statistically significant.

## 3. Results and Discussion

### 3.1. Surface Structure of the Porous Polyurethane Film Used in AlveoMPU

The flexibility, foam diameter, and foam density of foamed P-PU can be controlled using a pore size that allows cells to sufficiently extend within the pores while preventing easy translocation of cells to the opposite side of the membrane. A cross-sectional SEM image of the P-PU thin film used in this study is shown in Figure 2a and Appendix A. The side opposite to the pedestal PP plate used for membrane preparation is defined as the upper side of the membrane, while the side directly in contact with the PP plate is defined as the lower side of the membrane. The upper side of the membrane had larger pores, whereas the substrate-adjacent lower side featured relatively smaller pores, forming an anisotropic and interconnected micro pores. Owing to the foam properties, the pores displayed a vertical orientation with a gentle slope, forming a concave mortar-like structure downwards. Pore sizes were distributed, with larger ones on the upper side ranging from 4–13 µm (with a peak value of about 7 µm) and smaller ones on the lower side ranging from 1–4 µm (with a peak value of about 3 µm) as shown in Figure 2c. The pores generated by foaming were independently distributed, but there were minor connecting paths between two adjacent pores (Appendix A). The total thickness of the membrane, approximately 5 µm, was slightly thinner than the structure of commercially available cell culture insert membranes. Conversely, commercially available membranes, such as PET, typically have pores formed by electron beams, resulting in pores that are vertical and linear relative to the membrane surface (Appendix A). AFM observation revealed that the surface of the P-PU in AlveoMPU membrane exhibited microporous structures, dispersed on a smooth plane (Figure 2b). The mean Ra of the flat areas on the P-PU surface within the AlveoMPU membrane was calculated to be 1.95 ± 0.53 nm, indicating that these areas are quite smooth. It is anticipated that the smooth surface of the AlveoMPU membrane will not impede cell elongation, as previously suggested [24].

### 3.2. Hardness Measurement

To investigate the effects of membrane stability and substrate stiffness on cellular processes such as cell progression and differentiation during cell culture, we conducted tensile testing to assess the hardness of various materials, including P-PU as material for AlveoMPU, non-foamed flat polyurethane (F-PU), and commercially used PET and PC (both with a nominal pore diameter of 3 µm). Representative stress–strain curves for the four specimens of each material are shown in Figure 2d. Comparison of the average elongation at break and maximum stress revealed that it was 55.1% and 44.6 N/mm^2^ for F-PU, 10.8% and 19.4 N/mm^2^ for P-PU, 11.4% and 166.6 N/mm^2^ for PET, and 23.8% and 62.0 N/mm^2^ for PC. A comparison of Young’s moduli in the 1–3% elongation range showed that the value was 9.15 MPa for F-PU, 5.26 MPa for P-PU, 31.26 MPa for PET, and 16.34 MPa for PC. These results indicate that P-PU is approximately 1.7 times softer than F-PU, with its porous structure rendering it more prone to elongation and breakage. Additionally, P-PU is approximately 5.9 times and 3.1 times softer than the commercial PET and PC membranes, respectively. Considering that the hardness of scaffolding materials influences cell proliferation, spreading, and differentiation [25], and acknowledging the wide variation in Young’s moduli among biological tissues (e.g., brain: 1–4 kPa, heart: 10–15 kPa, cartilage: 1000–15,000 kPa, bone: 150,000–200,000 kPa) [26], the materials compared in this study deviate significantly from tissues other than cartilage or bone. Nevertheless, scaffolding materials that are closer in hardness to the actual cellular environment within the body are expected to facilitate differentiation and bottom-up remodeling of cells. Thus, among the substrates compared, P-PU used in AlveoMPU is the most suitable option [27]. P-PU, like the other films used for comparison, has a Young’s modulus that is significantly higher than that of soft biological tissues, meaning it is much stiffer than biological tissues. However, this mechanical property may offer several practical advantages. Specifically, the high tensile strength of these materials helps support cell layers and maintain the stability of cultured tissues in vitro. This is particularly useful for long-term cultures and applications involving mechanical stress. Additionally, using materials with high tensile strength makes it easier to handle cell sheets and reduces the risk of damage. This characteristic is crucial for practical applications such as future regenerative medicine and cell-sheet transplantation in tissue engineering.

### 3.3. Wettability of P-PU

The effects of material wettability on cell adhesion have been discussed, and materials with exceptionally low or high wettability affect cell adhesion [28,29,30]. Common cell culture substrates, such as bare polystyrene, are not inherently conducive to cell adhesion due to their low wettability and high water contact angle, and often require surface treatments, such as plasma processing [31]. In this study, the wettability of porous P-PU and its base material, F-PU were evaluated using dynamic contact angles. The initial contact angle of F-PU was approximately 70°, but decreased over time, reaching approximately 50° after 200 s. In contrast, the initial contact angle of P-PU exceeded 100° but decreased over time, matching that of F-PU after 500 s and then falling below that of F-PU thereafter. This reversal in wettability is not attributed to an increase in the bulk wettability of P-PU but rather to the absorption of water droplets through capillary condensation facilitated by its porous structure. This phenomenon is supported by the observation that the diameter of the water droplets applied to P-PU did not decrease over time; only the droplet height decreased [32] (details are discussed in the Appendix A). Thus, it is assumed that when the medium with suspended cells is applied, water is drawn into the pores of P-PU, promoting the active induction of suspended cells into these pores during cell seeding. According to Tamada et al., the optimal contact angle for cell culture scaffolding materials is approximately 40–70° [28]. Moreover, because the cell adhesion depends on the bulk material characteristics rather than the porosity, it is reasonable to discuss cell adhesion in relation to the wettability of F-PU. Both F-PU and P-PU allowed cells to reach the substrate surface sufficiently via gravitational settling 10 min after cell seeding, indicating favorable wettability parameters for cell adhesion at this point. Consequently, subsequent experiments were conducted using P-PU without additional cell adhesion-promoting treatments such as plasma processing.

### 3.4. Time Course Transition of Barrier Function in AlveoMPU

The temporal progression of trans-epithelial electrical resistance (TEER) as an indicator of barrier function, is shown in Figure 3a. Primary human lung microvascular endothelial cells (HLMVECs) were seeded on Day 0, and primary human alveolar epithelial cells (HAEpiCs) were seeded on Day 1, with TEER measurements commencing 24 h after Day 2. TEER increased over time, reaching approximately 470 Ω·cm^2^ by Day 8. Notably, when alveolar epithelial cells were seeded alone on P-PU, an increase in TEER was not observed. This suggests a lack of paracrine action mediated by growth factors secreted by HLMVECs, resulting in inadequate barrier function. Additionally, from Day 2 onwards, in the monolayer culture of the alveolar epithelium, the media on the well side was unified with alveolar epithelial cells with reduced growth factor supplementation as used in apical side for two-layer AlveoMPU model. 

TEER values for primary alveolar epithelial cells from humans and rats are approximately 2000 Ω·cm^2^, while comparatively lower values were observed in this study [33,34]. This discrepancy may be attributed the larger pore size of P-PU used in AlveoMPU, often exceeding 3 µm (Figure 2c), indicating that TEER formation may require some time to develop. Conversely, previous studies have used porous membranes with electron beam-drilled holes of less than 1 µm. However, Suresh demonstrated that H441 cells, a cell line of alveolar epithelial cells known to possess barrier function equivalent to primary human alveolar epithelial cells, have TEER values between 300–400 Ω·cm^2^ and the Papp of high-molecular-weight dextran showed equivalent values compared to those in humans and rats [35]. Generally, Papp is inversely proportional to TEER; samples with hundreds of TEER values tend to reach saturation in the permeation rate, resulting in equivalent levels. Thus, the TEER values obtained in the AlveoMPU model were expected to be sufficiently applicable for pharmacokinetic tests. 

### 3.5. Distribution of Alveolar Epithelial and Endothelial Cells on AlveoMPU

A comprehensive SEM image illustrating the adhesion of HAEpiC to P-PU of AlveoMPU is shown in Figure 3b. On P-PU, characterized by 4–7 µm large mortar-shaped structures, two distinct cell morphologies can be observed: the smooth type I alveolar epithelial cells (AT1) that occupy most of the alveoli and facilitate gas exchange, and the type II alveolar epithelial cells (AT2) with three-dimensional and microvillous structures that secrete surfactant and can serve as precursors to AT1. Despite minor contraction and detachment observed during SEM processing, AT1 cells smoothly expanded and covered the large pore structures on the upper side, bridging the large mortar-shaped pores (Figure 3(b-i)). Conversely, AT2 cells extend pseudopods into the pores, indicating expansion as well (Figure 3(b-ii)). Subsequent TEM images provided a deeper insight into the of ultra-thin sections of AlveoMPU, depicting a dual-layer sheet comprising HAEpiCs and HLMVECs (Figure 3(c-i)), with a close-up view revealing the proximity of both cell types to P-PU (Figure 3(c-ii)), and a close-up of the peripheral areas of HAEpiCs (Figure 3(c-iii)). HAEpiCs infiltrated into P-PU’s mortar-shaped pores, projecting toward the connecting pores and closely approaching the monolayer micro-thin HLMVECs sheets through the connecting pores. A detailed observation revealed both cell types closely juxtaposed within 1 µm, separated by a thin collagen film layer of approximately 200 nm introduced for coating and inhibiting cell migration to the opposite site through the connecting pores (Figure 3(c-ii), Appendix A, white arrow). Given that the thickness of the basement membrane between alveolar epithelial-capillary endothelial cells in vivo ranges from about 100 nm to 2 µm [36,37,38], the thickness of the artificially coated collagen layer on the bottom of the P-PU membrane in AlveoMPU appears to adequately reflect the biological structure.

The exchange of materials between the two cell sheets is assumed to occur via diffusion, governed by Fick’s second law [39]. According to this law, the concentration of a substance at a given point on a cell membrane, represented by *φ (r*, *t)*, is influenced by the distance from the source (*r*) and the elapsed time (*t*). This relationship is described by Equation (1):(1)$$\varphi \left(r,\;t\right)=\frac{1}{\sqrt{4\pi Dt}}\;exp(-\frac{r2}{4Dt})$$

Under constant conditions of the diffusion coefficient (*D*) and elapsed time, the concentration of substances between the two cell layers, if limited to passive diffusion, is exponentially influenced by distance, with the diffusion concentration exponentially increasing as the distance decreases. The porous membranes used in commercial cell culture inserts are approximately 10 µm thick, serving as the intercellular distance (Appendix A). Given that the intercellular distance for the two layers formed by P-PU in AlveoMPU is approximately 1 µm as observed in the TEM images (Figure 3(cii)), the ratio of thickness to P-PU compared to that of commercial porous membranes is exp(99/4Dt) times from Equation (1). Given that the diffusion coefficients of peptide hormones or cytokines in PBS are around 10^−6^ cm/s [40], the concentration ratio for hormones after one day (86,400 s) between both is an astounding exp (286.4583) (2.5 × 10^124^) times. Therefore, the closely apposed dual-layer sheet formed by AlveoMPU represents a unique in vitro alveolar model capable of evaluating cell–cell interactions not attainable in co-cultures separated by commercial cell culture inserts, reflecting a realistic intercellular distance found in vivo. 

Furthermore, the proximal areas of the collagen-coated layers and HAEpiCs revealed structures resembling hemidesmosomes near the basement membrane in the TEM images, identified by slightly darker contrast areas (Figure 3(cii) and Appendix A, black arrows). These hemidesmosomes, structures termed adhesion plaques that link integrin proteins on the cell surface with the extracellular matrix on the basement membrane, function to stably maintain alveolar epithelial cells within alveolar structures and are disrupted in lung diseases owing to pathogenesis or cytokines [41,42,43]. In Figure 3a, an increase in TEER of AlveoMPU over time was observed, which was supported by ultra-thin TEM sections. The imaged sections revealed staining with intermediate contrast, showing the structures of tight junctions, which provide the barrier function between cells, at the apical side of alveolar epithelial cells (Figure 3(ciii), TJ). Additionally, desmosomes, which strongly anchor cells together, were observed in proximity to the center of the cells with strong contrast in staining, both between and within cells (Figure 3(ciii), DS). These structures are observed in vivo [44], suggesting that AlveoMPU functions as a model closely resembling living tissue suggests the potential for developing models to assess the formation and disintegration of tight junction, desmosome and hemidesmosomes using specific antibodies or fusion fluorescent proteins.

### 3.6. Inflammation Induction and Therapeutic Effect Trials Using AlveoMPU

Pathogen-associated molecular patterns (PAMPs) are conserved molecular structures commonly found in bacteria, viruses, fungi, and parasites [45,46]. These molecular structures are key for the host immune system to specifically recognize pathogens and trigger immediate responses. PAMPs are recognized by receptors on the host cell surface, known as pattern recognition receptors (PRRs), which initiate innate immune responses, enabling the host to respond rapidly to infections, and promoting the elimination of pathogens. Toll-like receptors (TLRs) are PRRs that play crucial roles in recognizing pathogen components and activating the host’s innate immune response [47,48].

Poly (I:C), a synthetic analog of double-stranded RNA (dsRNA), mimics the dsRNA structure produced during viral replication process of many viruses. It is recognized as a ligand for toll-like receptor 3 (TLR3) and is used to induce immune responses that mimic viral infections [49]. Lipopolysaccharide (LPS), found in the outer membrane of Gram-negative bacteria, is a potent inducer of immune responses and is recognized as a primary ligand for toll-like receptor 4 (TLR4). The interaction between LPS and TLR4 triggers diverse immune responses to combat bacterial infections.

We investigated the feasibility of evaluating the production of cytokines (IL-6, IL-8, TNFα) involved in inflammatory responses by administering poly (I:C) and LPS from the apical side (corresponding to inner cavity side of the alveoli) using AlveoMPU, simulating respiratory infections.

ELISA data of cytokine secretion from cells following the administration of pathogen-mimicking substances poly (I:C) and LPS are shown in Figure 4a–i. To evaluate the inflammatory responses, media were collected not only from the apical side but also from the basolateral side (corresponding to the blood side within the capillaries) for performing ELISA. Additionally, data were compared with those obtained using the AlveoMPU bilayer cell model, including cultures of alveolar epithelial cells alone using P-PU and cultures of vascular endothelial cells alone in standard-well plates.

The results revealed no significant increase in IL-6 secretion on the apical side when poly (I:C) and LPS were administered alone; however, a statistically significant increase in secretion was observed only on the basolateral side under bilayer culture conditions of AlveoMPU (Figure 4a,b). Furthermore, treatment with poly (I:C) resulted in a significant increase in IL-8 secretion only on the apical side in alveolar epithelial cells. Conversely, IL-8 secretion strongly increased only under bilayer culture conditions of AlveoMPU on the basolateral side (Figure 4d,e). Similarly, TNF-α secretion did not significantly increase on the apical side following treatment with poly (I:C); however, a significant increase was observed on the basolateral side under bilayer culture conditions of AlveoMPU.

The increase in cytokine secretion induced by these mimetic substances was suppressed by pretreatment with fluticasone, a corticosteroid used in respiratory therapy, with no statistically significant difference observed compared to the controls (Figure 4a,b,d,e,g,h, AlveoMPU + Flu). These results indicated that cytokine secretion was induced by poly (I:C) and LPS in the AlveoMPU bilayer culture, validating the function of TLR3 and TLR4, thus proposing their utility as models for evaluating inflammatory responses, including analyzing downstream signals of PRRs. Additionally, the suppression of cytokine secretion by anti-inflammatory drugs, such as glucocorticoids, demonstrates their potential application to therapeutic drug screenings. Moreover, there were cases where cytokine secretion was promoted on the basolateral side under AlveoMPU bilayer culture conditions compared to alveolar epithelial or vascular endothelial cells alone, suggesting that reactivity to PAMPs was heightened because of the interaction between the two cell types forming a closely adjacent sheet. This indicates the potential use of AlveoMPU as a model for systemic inflammatory response syndrome, in which local inflammatory responses affect the systemic circulation.

To determine the feasibility of using AlveoMPU as a model for drug kinetics, the Papp of high-molecular-weight fluorescent dextran (4 kDa, FD-4) and fluticasone were determined by administering them to the apical side and collecting the media from the basolateral side 24 h later to measure their concentrations (Figure 4j). The Papp of FD-4 was approximately 1.4 × 10^−7^ cm/s, and that of the low-molecular-weight therapeutic drug fluticasone was 2.0 × 10^−6^ cm/s. The data for FD-4 were approximately one order of magnitude larger than previously reported values [33], suggesting that the permeation of large molecules, polymers, or particles greatly depends on the size of the semi-permeable membrane acting as the cell scaffold. Traditional models of alveolar epithelial cells use membranes with pore sizes less than 1 µm to limit translocation across the membrane, while AlveoMPU includes those with pore diameters larger than 3 µm and a much higher occupancy rate on the thin film surface of the pores, which may explain these discrepancies. Owing to the lack of data on the TEER values of fluticasone in primary alveolar epithelial cells in existing literature, direct comments on the validity of the data obtained in this study cannot be made. However, considering the equivalent rates for high- and low-permeability drugs in airway epithelial cell line Calu-3 and small-intestinal epithelial cell line Caco-2, and the permeability of steroid compounds testosterone and dexamethasone in both cells being approximately 3–12 × 10^−6^ cm/s [50], the AlveoMPU model can be considered valid as a predictive model for drug permeability in alveoli. 

In our study, the Papp of the high-molecular substance FD4 were assessed upon administration of pseudo-infectious agents poly (I:C) and LPS, both with and without fluticasone pretreatment, in both primary human alveolar epithelial cells (HAEpiCs) alone and within the AlveoMPU model (Figure 4k). The permeability rate in HAEpiCs alone exceeded 4 × 10^7^ cm/s, which was more than four-fold higher than that observed in the AlveoMPU model (*p* < 0.01, ANOVA). This supports the findings shown in Figure 3a, where the TEER values of the AlveoMPU model were more than four times higher than those of HAEpiCs alone. A trend of decreased permeability rates was observed in the AlveoMPU model with fluticasone pretreatment. Conversely, treatment with fluticasone followed by poly (I:C) and LPS showed an increase in the permeability rate compared to that of the control condition. Although these trends were not statistically significant, they may reflect the potential of the AlveoMPU to replicate the improvement in epithelial barrier function and the reduction in permeability seen with glucocorticoid administration [51], as well as the decrease in barrier function observed under inflammatory conditions in vivo [52].

We showcased the AlveoMPU model as a significant advancement in modeling the alveoli utilizing alveolar epithelial and capillary endothelial cells. This model replicates the complex nano-level architecture of the alveoli, including ultra-thin layers of alveolar epithelial and vascular endothelial cells positioned closely across a nano-sized basement membrane, along with structures like hemi-desmosomes and tight junctions. However, we acknowledge that the interactions with other cell types, such as alveolar macrophages and fibroblasts present in the basal membrane, were not fully considered in our initial model formulation [53]. Moving forward, it is crucial to introduce these cells into the AlveoMPU to achieve a replication of the in vivo environment more accurately. Furthermore, we aim to elucidate the differences in specific phenomena occurring between cells cultured on the AlveoMPU and those cultured on commercial membranes through comprehensive analytical techniques, such as transcriptomics, proteomics, and metabolomics, since this study only evaluated specific indicators including several inflammatory cytokines, barrier function, and pharmacokinetics. We believe that enhanced AlveoMPU models will underscore their potential as an effective alternative to animal testing in respiratory research and drug development, deepening our understanding of the complex interplay within the alveoli and contributing significantly to the field of respiratory research and medicine.

## 4. Conclusions

Through controlled foaming of polyurethane, we created a mortar-shaped polyurethane sheet featuring anisotropic pores on both sides of the thin film. By incorporating P-PU into cell culture inserts and seeding both alveolar epithelial and pulmonary microvascular endothelial cells, we established an alveolar model referred to as AlveoMPU. These bilayer cell sheets can interact closely with each other, making it possible to replicate an intricate alveolar microenvironment that cannot be captured with traditional cell culture inserts. AlveoMPU is applicable to lung inflammation models and may be used for evaluating anti-inflammatory drugs and as a model for pharmacokinetics. AlveoMPU offers a platform that contributes to the understanding of lung diseases and screening of therapeutic drugs, potentially replacing animal-based research methods and accelerating the pace of scientific discovery and therapeutic development in the field of pulmonary medicine.

## Figures and Tables

**Figure 1 polymers-16-01486-f001:**
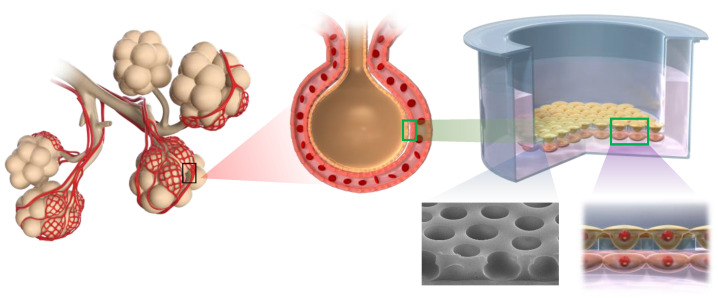
Conceptual diagram of AlveoMPU simulating the actual structure of the lungs and the interaction between alveolar epithelial cells and microvascular endothelial cells.

**Figure 2 polymers-16-01486-f002:**
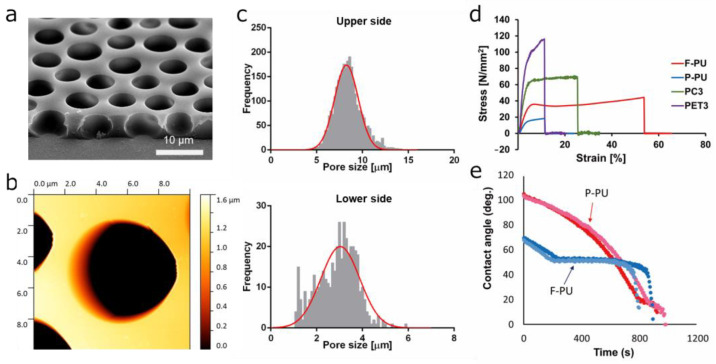
Characterization of the physicochemical properties of the porous polyurethane membrane (P-PU) used in AlveoMPU. (**a**) Scanning electron microscopy (SEM) image viewed from an oblique angle. (**b**) Atomic force microscopy (AFM) height image of the AlveoMPU membrane. (**c**) Distribution of pore diameters for the upper and lower sides of the AlveoMPU membrane. Red curved line: Gaussian distribution obtained from the frequency data of pore sizes. (**d**) Stress–strain curve for flat-polyurethane (F-PU), porous PU (P-PU; AlveoMPU), polycarbonate with 3 µm pores (PC3) and polyethylene terephthalate with 3 µm pores (PET3). (**e**) Time dependency of contact angles of F-PU and the P-PU. Two independent samples for F-PU and P-PU were replicated and are denoted by the blue and red curved lines, respectively.

**Figure 3 polymers-16-01486-f003:**
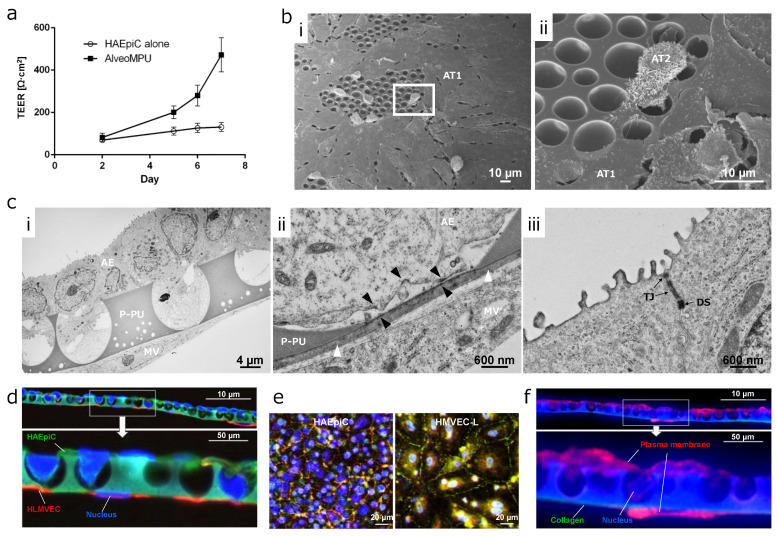
Formation of alveolar model and barrier function on AlveoMPU membrane. (**a**) The temporal progression of trans-epithelial electrical resistance (TEER). (**b**) Scanning electron microscopy (SEM) images of the top surface of AlveoMPU; (**b**-ii): zoomed-in view of the image in (**b**-i); AT1 and AT2: type 1 and 2 alveolar epithelial cells, respectively. (**c**) Transmission electron microscopy (TEM) images of a cross section of AlveoMPU; P-PU: porous polyurethane of AlveoMPU, AE: alveolar epithelial cell, MV: microvascular endothelial cell; (**c**-i): The formation of sheets by alveolar epithelial and vascular endothelial cells across P-PU; (**c**-ii,**c**-iii): magnified views within white and black frames in (**c**-i); white arrowhead: collagen thin layer, black arrowhead: hemidesmosome-like structure, TJ: tight junction, DS: desmosome. (**d**) Immunostaining of alveolar epithelium (green: cytokeratine A1/A3) and vascular endothelium (red: vimentin)-specific markers and (**e**) tight junction marker proteins (green: Occludin, red: ZO-1, blue: nuclei) and (**f**) collagen backing layer (green: Type I Collagen, red: plasma membrane).

**Figure 4 polymers-16-01486-f004:**
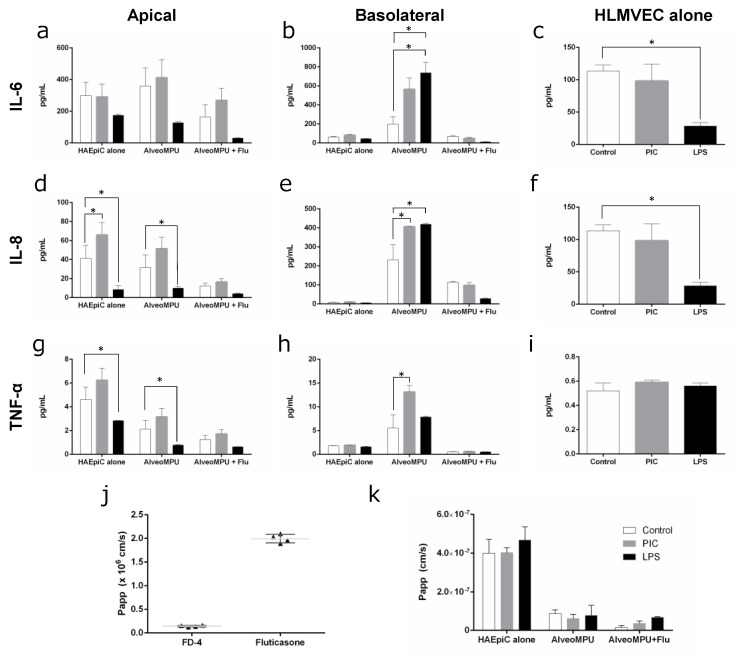
Evaluation of inflammatory cytokine secretion and pharmacokinetic testing utilizing AlveoMPU. Cytokine production of AlveoMPU models into apical (**a**,**d**,**g**), basolateral sides (**b**,**e**,**h**) or HLMVECs culture supernatant (**c**,**f**,**i**); IL-6: (**a**–**c**), IL-8: (**d**–**f**) and TNFα: (**g**–**i**). (**j**) Apparent permeability coefficient (Papp) of 4 kDa dextran-FITC (FD4) and fluticasone propionate. (**k**) Papp of FD4 after pretreatment of fluticasone propionate; white bar: control; gray bar: poly (I:C) (PIC); black bar: lipopolysaccharide (LPS); *: statistically significant difference (*p* < 0.05).

## Data Availability

Datasets are contained within the article and Appendix A.

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
