# Peer review of "AlveoMPU: Bridging the Gap in Lung Model Interactions Using a Novel Alveolar Bilayer Film"

_polymers, 2024, doi:10.3390/polym16111486_

Round 1

Reviewer 1 Report

Comments and Suggestions for Authors

The current manuscript entitled "AlveoMPU: Bridging the Gap in Lung Model Interactions Using a novel alveolar bilayer film" focuses on establishing alveolar bilayer film that facilitates direct cell-cell interaction. The manuscript was clearly written, however few minor revisions should be made before further process.

1. Add more informations/discussion about other lung models used in vitro, and the comparison with alveolar bilayer film.

2. include Advantages of  characteristics of bilayer film like porosity, tensile strength and its relation with lung delivery??

3. Add future perspective of the current study.

Comments on the Quality of English Language

English language slightly needs improvement.

Reviewer 2 Report

Comments and Suggestions for Authors

In this manuscript, the authors develop a novel strategy for designing a polymer-based material that emulates alveolar epithelial cells. To create the porous framework, the authors utilize the chemical foaming of polyurethane. They extensively analyze the material's surface morphology and physical traits to gauge its quality. The manuscript's objective is promising, offering broad applicability for designing pulmonary epithelial cells for drug testing and studying pharmacokinetics. The manuscript is overall well-written and addresses the strengths and downsides of the method which is quite useful for the readers. I recommend the publication of the paper, provided the authors address the questions raised below:

  1. The authors do not comment on the potential toxic response of the material and how that can hinder its applicability for practical purposes. A paragraph on that as a part of the discussion section or conclusion would warrant the readers regarding how the material can be utilized further for specific responsive behavior. 
  2. The Young’s moduli of F-PU and P-PU are substantially different than any biological tissue as reported here. What would be the implication of this difference in tensile strength on the practical application of the material? It would be nice if the authors addressed the issue in the manuscript in more detail. 
  3. I am a bit concerned about how the contact angle and therefore the wetting capability of P-PU changes with time. While the rate of decrease is slower than the base material and is attributed to the porosity of the material, it would be useful to know how the decrease in wetting might affect the cell adhesion behavior in a quantitative manner. 
Comments on the Quality of English Language

The english seems to be fine and clear
